# Experimental Implications of Negative Quantum Conditional Entropy—H₂ Mobility in Nanoporous Materials

**C. Aris Chatzidimitriou-Dreismann**

Institute of Chemistry, Sekr. C2, Faculty II, Technical University of Berlin, D-10623 Berlin, Germany; chariton.dreismann@tu-berlin.de; Tel.: +49-30-314-22692

**Abstract:** During the last few decades, considerable advances in quantum information theory have shown deep existing connections between quantum correlation effects (like entanglement and quantum discord) and thermodynamics. Here the concept of conditional entropy plays a considerable role. In contrast to the classical case, quantum conditional entropy can take negative values. This counter-intuitive feature, already well understood in the context of information theory, was recently shown theoretically to also have a physical meaning in quantum thermodynamics [del Rio et al. Nature **2011**, 474, 61]. Extending this existing work, here we provide evidence of the significance of negative conditional entropy in a concrete experimental context: Incoherent Neutron Scattering (INS) from protons of H₂ in nano-scale environments; e.g., in INS from H₂ in C-nanotubes, the data of the H₂ translational motion along the nanotube axis seems to show that the neutron apparently scatters from a fictitious particle with mass of 0.64 atomic mass units (a.m.u.)—instead of the value of 2 a.m.u. as conventionally expected. An independent second experiment confirms this finding. However, taking into account the possible negativity of conditional entropy, we explain that this effect has a natural interpretation in terms of quantum thermodynamics. Moreover, it is intrinsically related to the number of qubits capturing the interaction of the two quantum systems H₂ and C-nanotube. The considered effect may have technological applications (e.g., in H-storage materials and fuel cells).

**Keywords:** quantum thermodynamics; quantum information; negative conditional entropy; quantum correlations in condensed matter; nanoscale quantum confinement; proton quantum mobility; incoherent neutron scattering; H storage materials; fuel cells; quantum materials

---

## 1. Introduction: "Information is Physical"

During the last few decates, theoretical advances in (non-relativistic) quantum mechanics have provided novel insights into several physical and technological fields, and have also allowed us to predict new effects and to invent new technological applications. In particular, various quantum correlation effects (e.g., Entanglement [1,2] and the newly discovered Quantum Discord [2–4]), being succinctly characteristic features of quantum mechanics, play a significant role here. Note that quantum coherence and quantum correlations, in particular entanglement, are also the basic ingredients of Quantum Information Theory (QIT); see [2,5]. Moreover, also in the context of classical information theory holds, as Landauer puts is: "Information is Physical" [6].

The focus of this paper is on a new scattering effect accompanying elementary collisions between two quantum systems—a neutron and a proton (or H atom)—in which H interacts with a non-classical environment, say $\mathcal{E}$. In contrast to the classical case, a quantum $\mathcal{E}$ will be shown to cause striking effects of energy and momentum transfers. Evidence of a recently discovered "anomalous" scattering

effect is presented and discussed—excess energy transfer, (equivalently: momentum-transfer deficit, or reduced effective mass of the scatterer).

In the examples under consideration below, the well-known technique of incoherent inelastic neutron scattering (INS) of thermal neutrons is applied. Note that INS clearly belongs to non-relativistic physics, and that the scattering is incoherent [7,8]; see also Appendix A. In an example, we consider single $H_2$ molecules in multi-walled carbon nanotubes, where H-neutron collisions result in translation along the nanotube axis. Interpreting the INS data with conventional theory [8–11], the measurements indicate a striking mass deficit of $H_2$: it appears to be only $M = 0.64$ atomic mass units (a.m.u.)—in blatant contradiction to its well known value $M = 2$ a.m.u.

To reveal the specific reason of the failure of the conventional view, it is necessary to consider certain important experimental aspects of the neutron scattering spectrometer and the concrete measuring procedure in more detail. Indeed, it is necessary to be aware of the following questions: What is, in fact, directly measured? Moreover, what is theoretically deduced from the experimental data, and how?

The aforementioned effect is exploited and interpreted "from first principles" in Sections 5–8, which constitute the theoretical part of the paper. The experimental INS results find a direct qualitative interpretation in the frame of modern quantum thermodynamics (QTD) and quantum information theory (QIT). Here, the concepts of entanglement, quantum discord and quantum conditional entropy appear to be essential.

A particular feature of the theoretical treatment is that it gives us a physical picture based on quantum correlations, rather than on quantum mechanisms.

To facilitate understanding, the following aspects concerning quantum correlations should be pointed out. Quantum correlations of electrons, being caused by the indistinguishability of identical particles, are already well known in traditional physics and chemistry of atoms, molecules and solids. These are not considered in this paper. In contrast, quantum correlations between non-identical particles (e.g., protons and neutrons) are all but unknown in most fields, like, e.g., chemistry, molecular biology and applied material sciences, but they play a central role here. They are also unknown in standard non-relativistic neutron scattering theory [8–11].

The paper's theoretical part is based on concepts of QIT [5] and QTD. Our analysis makes contact with studies of quantum entanglement and its extension and/or generalization within the theoretical framework known as quantumness of correlations, in short QoC. In particular, our particular interest is on experimentally accessible energetic and thermodynamical consequences of quantum correlations, as, e.g., the work costs for erasing quantum information. In specific quantum cases characterized by negative conditional entropy [12], these costs are predicted to be even negative [13]. The field of QoC is already very broad; see, e.g., [14] and references therein. But until now it remained rather unnoticed in most experimental fields (with exception of quantum optics).

The focus of this paper is on the experimental relevance of general theory for concrete scattering experiments—interpretational issues concerning basic quantum entities and/or purely theoretical derivations play only a minor role here. In other words, we concentrate on the connections between modern theory and concrete experimental fields, and propose a new group of effects showing "anomalous" energy and/or momentum transfers, having in mind that they may also have technological applications.

Specifically, we investigate the aforementioned counter-intuitive INS results from $H_2$ single molecules in C-nanotubes (and another nano-porous metal-organic material), as obtained with the aid of a new-generation two-dimensional spectrometer (like, e.g., ARCS [15] or MARI [16]). We present a natural, straightforward theoretical interpretation in the theoretical frame of QTD and QIT. Consequently, this demonstrates that these quantum-theoretical frameworks are not only very active areas of present-day research, but they also provide new conceptual ideas and techniques for various experimental (and perhaps also technological) applications.

More details of general CoQ theory underlying our investigations, of the theoretical derivations, and further experimental results, may be found in the original reference [17] and the most recent review article [18].

## 2. The Elitzur-Vaidman Effect, Google's Quantum Processor, and the Physical Meaning of the Quantum State Vector

One overarching open question of the interpretation of quantum physics—the "meaning" of the wavefunction, see [19] for a recent review—appears in new light after recent progress achieved through experiments with single probe systems (particles or photons).

The possibility that the wavefunction (or, the state vector) is an epistemological entity, i.e., solely a mathematical quantity representing the observer's knowledge, is often associated with the so-called "orthodox" interpretation of quantum mechanics. An ingredient of this is to consider the state vector to be associated with an ensemble of quantum systems, but not with individual ones.

The alternative viewpoint (which emerged with the aid of new experimental techniques using single quantum particles or photons) is to regard the state vector as an ontological entity, or a "real" physical entity. Clearly, this viewpoint was unacceptable for the overwhelming majority of physicists in the last century.

In this context, we may emphasize the significance of the counter-intuitive Elitzur-Vaidman effect [20] concerning interaction-free measurements (IFM, popularly also known as "bomb tester"), which revealed the ability to obtain experimental information about an object's presence in some position of one path of a Mach-Zehnder interferometer (MZI), without ever "touching" it. In the conceptual frame of classical physics, this is paradoxical. Namely, in successful interaction-free "bomb detections", absolutely no physical quantity—e.g., energy, momentum, spin, force, etc.—has been exchanged between the object and the probe particle. However, this physical information (here: bomb detection) cannot be gained "at no costs"; the associated "costs" are provided by the photon's (particle's) wavefunction, following the principles of IFM. Therefore, the experimental verification of this novel quantum effect (with photons and neutrons) demonstrated that the wavefunction represents a real physical quantity, rather than an auxiliary mathematical construct for the calculation of expectation values of quantum observables, or transition probabilities.

The interpretation of the quantum state vector has recently become relevant to practical applications such as quantum cryptography and quantum computation; see, e.g., the achievement to demonstrate experimentally the so-called "quantum supremacy" offered by quantum computation (as compared with classical computation)—using Google's quantum processor Sycamore, which has 53 (solid-state superconducting) qubits, and executes a specific computational task [21]. This work is reporting a (very specific) quantum computational task, performed in about 200 s—in contrast to the estimated 10,000 years needed by the presently existing most powerful classical supercomputer to perform the same task. (For another processor using quantum-optical elements, see reference [22].) To stress this point, one may notice the following. The specific state vector resulting from a calculation with the Google processor is a quantity referring to a single processor, and certainly not to an ensemble of quantum processors!

Parenthetically, it may be interesting to notice that the existence of the aforementioned quantum supremacy in the sense of basic computation complexity theory [23,24] is still an unknown issue. This holds because until now no quantum algorithm (or mathematical proof) has been found, which would demonstrate the efficient (i.e., polynomial-time) quantum-theoretical solution of any known NP-complete problem [23,24]. Indeed, many computer-science theoreticians believe that such a quantum supremacy does not exist at all—i.e., that QIT does not provide fundamentally more computational power than classical IT.

### 3. INS Measurement—Outline of Some Results

Aiming at making the presentation self-contained, in Appendix A are presented several technical details of an INS spectrometer and a concise description of "What is Measured" in an INS experiment. Furthermore, this section deals with some results of it, as derived from the data with conventional theory, see, e.g., [8–11].

#### 3.1. Conventional Theory: Momentum and Energy Conservation in Two-Body Collisions

Let us consider in some detail a neutron colliding with an atom (nucleus) of mass $M$. We use the notations defined in Appendix A. The atomic initial momentum $\mathbf{P}$, which in general is not zero, causes the following energy transfer $E = E_0 - E_1 \equiv \hbar\omega$ being transferred from the neutron to the atom ($E_0$: neutron's initial energy; $E_1$: neutron's final energy; see Figure A1):

$$
\begin{aligned}
E = E_0 - E_1 \equiv \hbar\omega &= \frac{(\hbar\mathbf{K} + \mathbf{P})^2}{2M} - \frac{\mathbf{P}^2}{2M} \\
&= \frac{(\hbar\mathbf{K})^2}{2M} + \frac{\hbar\mathbf{K} \cdot \mathbf{P}}{M}
\end{aligned}
\tag{1}
$$

This relation represents the conventional energy conservation. $\hbar\mathbf{K}$ is the momentum transfer from the neutron to the atom. In the following we use the common notation $|\mathbf{X}| = X$, for any vector $\mathbf{X}$.

The first term in the right-hand side (rhs) of the last equation is often called recoil energy,

$$
E_{rec} = \hbar\omega_{rec} = \frac{(\hbar K)^2}{2M}
\tag{2}
$$

It gives the gained kinetic energy of a struck atom being initially at rest. Hence, since it holds $\langle P \rangle = 0$ before collision, we obtain for the involved mean values the relation

$$
\langle E \rangle = \frac{\hbar^2 \langle K \rangle^2}{2M} \equiv E_{rec}
\tag{3}
$$

This equation refers to the center of the measured peak. Hence, INS from a gaseous sample of atoms leads to an intensity peak centered at energy transfer $E_{rec}$, and having a finite width being caused by the additional term $\hbar\mathbf{K} \cdot \mathbf{P}/M$. This term represents the so-called Doppler broadening (see also Appendix A). Peak center and peak width are illustrated in Figure 1, which shows data of incoherent neutron scattering from liquid helium, $^4$He; for details see [25].

The above simple formulas of this section capture the features of the so-called Impulse Approximation (IA) [10,26]. Obviously, the IA describes ultra-fast two-body collisions.

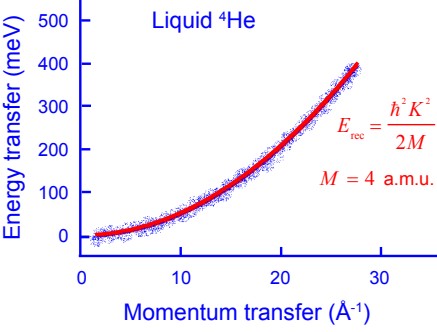

**Figure 1.** Schematic representation (blue points) of measured dynamic structure factor $S(K, E)$ of liquid helium [25]. The red line is the calculated recoil parabola, for the mass of $^4$He, shown as a guide to the eye. The white-blue ribbon around the recoil parabola represent data points measured with the time-of-flight spectrometer ARCS [15] (Figure taken from reference [26] with permission of Quanta).

### 3.2. Experimental Determination of Scatterer's Mass

Deviations from the impulse approximation (IA) are often observed in experiments; see, e.g., [10,26]. In the frame of conventional theory, such deviations can be understood as follows.

Energy conservation (1) for a two-body collision holds in the IA, the latter being exactly valid at very large momentum transfers and, quivalently, for scattering off quasi-free particles. For finite energy and momentum transfers of most actual experiments, however, the IA is not completely fulfilled, and hence so-called final-state effects (FSE) become observable [10,26]. FSE are due to interactions of the struck particle with its environment, which affect both initial and final states of it. In this subsection we shortly explain the conventional FSE.

When a particle with mass $M$ is not completely free but bound (even weakly) to other adjacent particles (in short: environment), the incoming neutron effectively scatters from a particle having effectively a higher mass than $M$. This effect is because the adjacent particles are exerting forces on the scattering particle, the free motion of which can only be hindered by these forces. This can only lead to and increase—and never a decrease—of its effective mass $M_{eff}$

$$M_{eff} \geq M \equiv M_{free}. \tag{4}$$

This reasoning is well understood; see references in [10,26].

This inequality can also be derived from the aforementioned energy conservation relation, if one includes an additional term $E_{int} > 0$ caused by the atom-environment interactions (or forces):

$$\bar{E} = \frac{\hbar^2 \bar{K}^2}{2M} + E_{int}, \tag{5}$$

$\bar{E}$ and $\bar{K}$ refer to the center of a measured peak. Hence, to the recoiling particle, there will be a reduced amount of energy, $\bar{E} - E_{int}$, available as its kinetic energy after the collision. Obviously, $\bar{E}$ and $\bar{K}$ are determined from the neutron's kinematics, in contrast to $E_{int}$, which is a quantity of the sample's material (i.e., of the scatterer). Now one can try to fulfill this equation using a new pair of energy and momentum transfer variables $(E_{IA}, K_{IA})$, and applying the conventional theory in the IA. This yields

$$E_{IA} = \frac{\hbar^2 K_{IA}^2}{2M_{eff}}. \tag{6}$$

Obviously, if one wants to describe the collision as a two-body process, and fulfill the equation $E_{IA} = \bar{E}$, the last two equations yield

$$M_{eff} > M \tag{7}$$

Additionally, this effect has been demonstrated by Watson, who showed its existence with the aid of an exact calculation of neutron scattering from a harmonic oscillator; see Figure 2 of reference [10].

For more examples, references and associated discussions of "effective mass" and FSE, the interested reader may consult [10,18,26].

### 3.3. INS from Bulk Ice-Ih—Conventional Theory

For illustration of the concepts and relations of the preceding subsection, we consider here a concrete experimental example: INS from H of $H_2O$ molecules of ice .

Usually, INS from H-containing samples is done at very low temperatures, because this reduces the "background" and Debye-Waller-factor effects [8] on the measured $S(K, E)$. The latter is the quantity of primary interest for quantifying material properties and testing theoretical predictions.

Liquid and solid water (ice) have been studied extensively in the INS scientific literature. Figure 2 shows the experimentally observed $S(K, E)$ of micro-crystalline ice Ih at $T = 20$ Kelvin and 1 bar pressure. Neutron's initial energy was $E_0 = 750$ meV. The measurements were done with the TOF

spectrometer MARI [16]. The most significant scattering contributions are caused by H, which has a large incoherent scattering cross-section [8]. The black parabola line represents the conventional recoil line of a free H being initially at rest

$$E_{rec}^{free} = \frac{\hbar^2 K^2}{2M_H},$$

(8)

($M_H$: atomic mass of H. Recall that H and neutron ($n$) have almost equal masses, $m_n \approx M_H$). Clearly, this is equivalent to Equation (3) above.

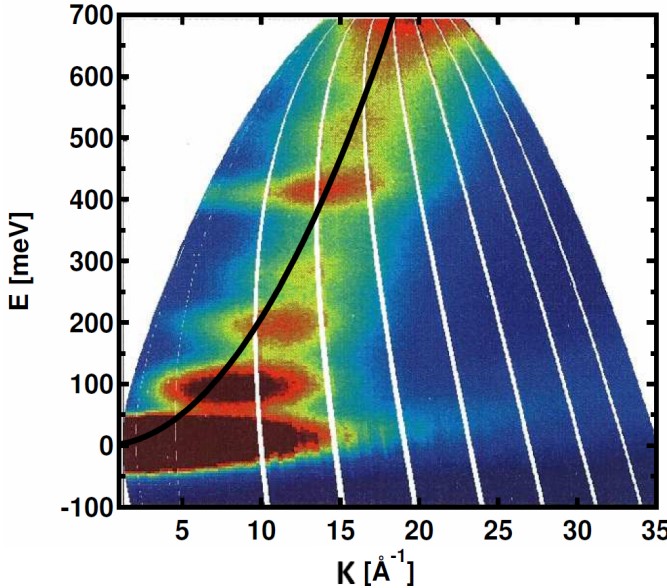

**Figure 2.** Experimental $S(K, E)$ intensity map of ice Ih at 20 K and 1 bar, recorded on MARI [16]. $E_0 = 750$ meV. The shown black line (parabola) is the conventional recoil trajectory of a free H. The strong intensity ribbon at $E \approx 0$ is mainly due to the aluminum cell containing the water (ice). The intensity peak centered at $E \approx 420$ meV and $K \approx 14$ Å$^{-1}$ represents the vibrational stretching OH-vibrations. See the discussion in the text. (Reproduced from reference [17] with permission.)

The intensity peak around $E = 420$ meV and $K = 14$ Å$^{-1}$ is due to the molecular stretching vibrational modes of $H_2O$. Here one observes that the center of intensity peak lies on the free H-atom recoil line.

In view of Equation (8) and relation $m_n \approx M_H$, this experimental result shows that the observed intensity peak is well described with the common theoretical assumption that the collision is impulsive and essentially a two-body (i.e., neutron-H) scattering process., i.e., there is neither a significant free recoil of the $H_2O$ molecule, nor a "dressing" of H with environmental and/or intramolecular degrees of freedom. Thus the effective mass of H is here $M_H^{eff} = M_H^{free} \approx 1.0$ a.m.u.

The following physical insight provided by Kearley et al. [27] gives additional justification to the above physical picture. Namely, the very short (being in the pico- to femto-second time range) neutron-H scattering time, taken together with the significant spectral separation of the stretching modes from the other vibrational modes, effectuate a decoupling of the "fast" H-stretching motion from the other (slower) H-motions.

We now consider the bending vibrational mode, i.e., the peak at energy transfer $E \approx 200$ meV. Interestingly, this peak does not lie on the free H-recoil parabola, but its central $K$-value appears at lower $E$-values. (Equivalently, it lies at higher $K$-values, at its central $E$-value). i.e., this peak exhibits a negative $E$-shift (or, equivalently, a "positive" $K$-shift).

However, also this result has a conventional explanation: This bending motion is hindered by adjacent H-bonded molecules, and thus the scattering H may be "dressed" by the aforementioned

environmental degrees of freedom. Thus this effect leads to an increased effective mass of the struck H. Indeed this is here observed: An estimate derived from the peak-center position, as shown in Figure 2, yields for the effective mass of the struck H: $M_H^{eff} \approx 1.3$ a.m.u. Thus we may conclude that, in contrast to the stretching modes, the bending vibration, having a lower frequency, is not fully decoupled from the other dynamical modes (i.e., librations, phonons, etc.) of the complex H-bonded material.

Summarizing, we have found that these experimental observations on ice represent no surprise at all, since they have a simple conventional-theoretical interpretation [8,11,17].

In contrast, in the case of $H_2O$ molecules confined in nano-scale channels, the stretching vibrational mode of $H_2O$ exhibits a large anomaly (with respect to conventional theory), as originally shown in [17].

## 4. The New Scattering Effect

### 4.1. Example: INS from Single $H_2$ Molecules in C-Nanotubes

Carrying out materials science research, a striking INS finding was recently observed [28] in the quantum excitation spectrum of $H_2$ adsorbed in multi-walled nanoporous carbon. The latter have pore diameters in the range 8–20 Å).

These experiments were carried out with the TOF spectrometer ARCS (SNS, O.R.N.L., USA) [15]. The temperature was $T = 23$ K. The incident neutron energy was $E_0 = 90$ meV. Consequently, the energy transfers in this experiment cannot excite $H_2$-vibrations, but rotations and translation (which is also called recoil) only. Note that $H_2$ interacts only weakly with the substrate. Thus it holds

$$E = E_{rot} + E_{trans}. \tag{9}$$

The measured two-dimensional INS intensity map $S(K, E)$ of H is presented in Figure 3, which is reproduced from the original publication of Olsen et al. [28]. The following features should be observed.The intensive peak centered at $E_{rot} \approx 14.7$ meV is due to the first rotational excitation $J = 0 \rightarrow 1$ of the $H_2$ molecule [11]. Furthermore, the corresponding momentum transfer of this intensity peak is $K_{rot} \approx 2.7$ Å$^{-1}$. Hence, the peak position in the $K - E$ plane fulfills the relation $E_{rot} = (\hbar K_{rot})^2 / 2M_H$ with the mass $M_H$ of the free H atom,

$$H_2 \text{ rotation } (J = 0 \rightarrow 1): \quad M_{eff} = M_H = 1.0079 \text{ a.m.u.} \tag{10}$$

Thus the spectral position of this rotational excitation in the $K - E$ plane is fully in line with conventional expectations, and one can say that each neutron scatters from a single H atom. Recall that an agreement with conventional theory was also observed in scattering from $^4$He.

However, the novel finding of [28] is the following. A detailed numerical analysis of the roto-recoil data (as shown in Figure 3) was presented, from which the authors extracted a strongly reduced effective mass of the recoiling $H_2$ molecule (left parabola, green):

$$H_2 \text{ translation (recoil): } \quad M_{eff}(H_2) \approx 0.64 \pm 0.07 \text{ a.m.u.} \tag{11}$$

Additionally, no significant sample-dependent variation in this value was found. This result is in blatant contrast to the conventionally expected value $M(H_2) = 2.01$ a.m.u. for a freely recoiling $H_2$ molecule (right parabola).

This striking experimental result provides strong evidence for the new "anomalous" effect of excess $E$-transfer (or equivalently, $K$-transfer deficit [26]) accompanying an elementary collision of a neutron with a recoiling molecule.

As already pointed out, every conventional $H_2$/substrate binding can never decrease the molecule's effective mass. Therefore these INS results contradict conventional theoretical expectations. However, as discussed below, they find a natural interpretation in the frame of QTD.

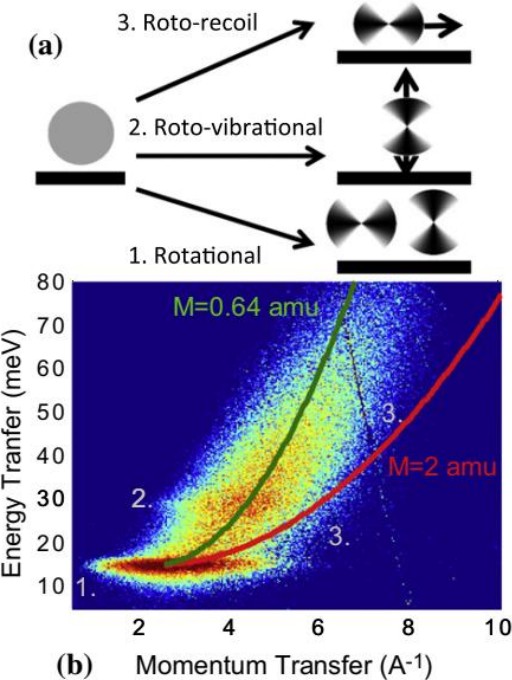

**Figure 3.** (**a**) Cartoon of various molecular motions of $H_2$ in a nanotube, as conventionally expected. (**b**) Experimental Incoherent Neutron Scattering (INS) results from $H_2$ in carbon nanotubes. Incident neutron energy: $E_0 = 90$ meV; Figure 1 of [28]. The translation motion of the recoiling $H_2$ molecules causes the observed continuum of intensity, usually called "roto-recoil" (yellow-orange ribbon). The $K - E$ position of the first rotational excitation of $H_2$ (dark red-brown ellipsoid). agrees with conventional theoretical expectations. In clear contrast, a detailed fit (green parabola) to the roto-recoil data yields the effective mass of translating $H_2$ to be $M_{eff} \approx 0.64$ a.m.u. The red parabola on the right shows the conventional-theoretical parabola associated with the $H_2$-mass $M = 2$ a.m.u. See the text for more explanations. (Reproduced from [28], with permission of Elsevier Ltd.).

### 4.2. Second Example—INS from Single $H_2$ Molecules in the Metal-Organic Framework Material "HKUST-1"

Due to its considerable magnitude, the surprising E-excess quantum effect discussed in the preceding subsection could be suspected to be an experimental artifact (e.g., due to wrongly calibrated spectrometer, erroneous data analysis, etc.). Hence it is important to have another, fully independent confirmation of this effect.

Aiming at discovering novel materials for potential use in the storage and transport of hydrogen for energy applications, also the adsorption of $H_2$ on porous metal–organic framework materials has attracted much attention over the past few years.

Among the wide diversity of such suitable materials, the material known as HKUST-1 (see [29] for its chemical formula and structure) has emerged as a model system for adsorption studies. For our present purposes, it is sufficient to note that HKUST-1 constitutes a three-dimensionally connected network that has a trimodal pore structure, having pores with diameters in the range 5–12 Å.

Figure 4 shows the measured intensity map $S(Q, E)$—$Q$ denotes in [29] the momentum transfer. The temperature was lower than 30 Kelvin [29]. Also here the data shows a broad band attributed to the translational $H_2$-motion in the pores, similarly to the experiment in the preceding subsection.

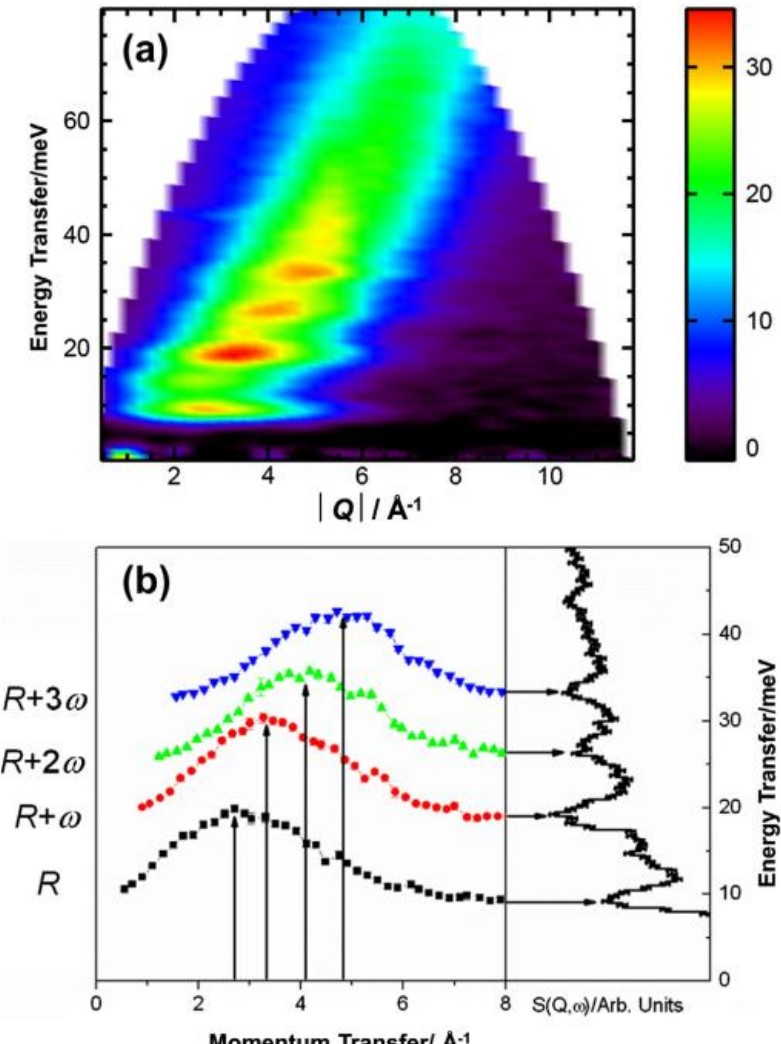

**Figure 4.** (**a**): The $S(Q, E)$ map of hydrogen for 1(p-H2):Cu in HKUST-1 for an incident energy of 75 meV measured using the MARI spectrometer. The green broad intensity ribbon is due to the translational quantum dynamics of $H_2$ and corresponds to the associated broad intensity ribbon shown in the preceding Figure 3. (**b**): Series of cuts along momentum transfer at a series of energy transfers corresponding to the intensity peaks shown in the right panel, i.e., the integrated (over momentum transfer) intensity for along energy transfer for the $S(Q, E)$ map shown in panel (**a**). (Reproduced from [29], with permission of Elsevier Ltd.).

### 4.3. Comparison of the Two Experiments

Very recently we became aware that this experiment [29] provides an independent confirmation of the striking result of reference [28]. The INS measurements were done with the 2-dim. spectrometer MARI (ISIS facility, Didcot, UK) by an independent European group of scientists (R. I. Walton and collaborators).

Visual comparison of the data of the two experiments reveal a very interesting result, see Figure 5. The two broad continuous bands due to the translational mode are virtually identically positioned in the $K - E$ plane. This is quite astonishing given the fact that the two experiments were done with two different nano-structured materials and using two different 2-dimensional spectrometers.

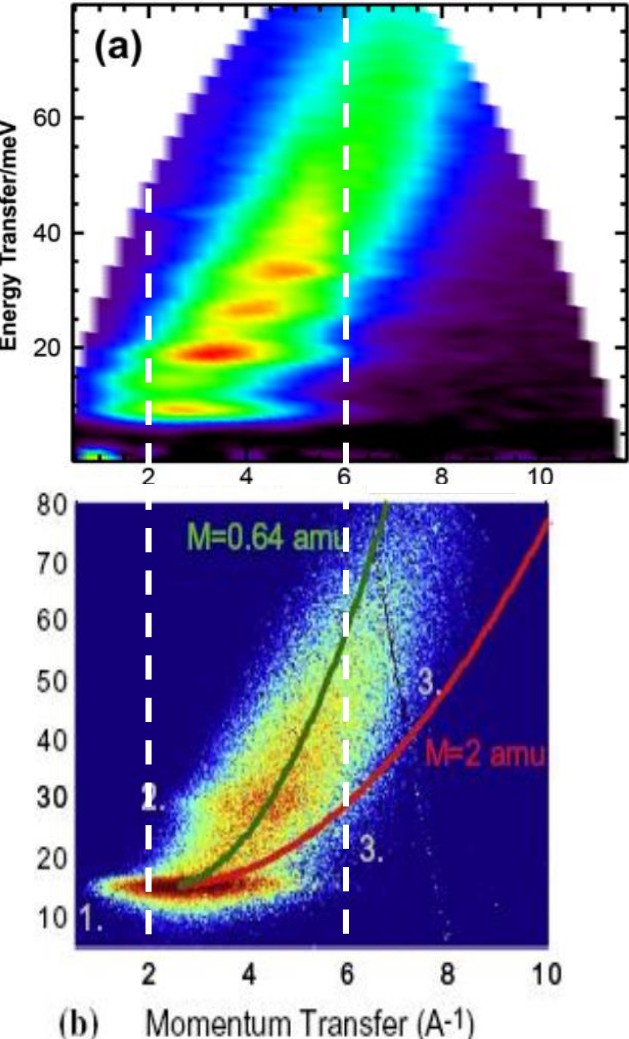

**Figure 5.** Comparison of the two experimental results: Experimental $S(K, E)$ maps of the two original papers, appropriately stretched/scaled to facilitate comparison. (**a**): data from [29]; (**b**) data from [28]. The shown two vertical dashed lines (white) are guides to the eye. The two broad continuous ribbons due to the translational modes are virtually identically positioned in the momentum-energy plane $K - E$.

In the light of these observations, the "anomalous" excess E-transfer (or, equivalently, momentum transfer deficit) effect under consideration appears to represent an established experimental result.

## 5. Quantum Heating and Quantum Cooling Due to Quantum Environment—Kurizki's Model

A theoretical model addressing explicitly the erasure of system-environment quantum correlations with a specific Hamiltonian mechanism, were proposed by Erez et al. [30] and Gordon et al. [31]. These investigations explore the erasure of correlations in the physical context of quantum cooling (and heating) of a two-level system undergoing frequent quantum non-demolition (QND) measurements [32]. Unexpected non-Markovian cooling/heating effects were shown to be related with novel features of coherent dynamics of the interaction between system and bath, which may take place during in times significantly shorter than the memory time of the bath's dynamics. This result is also in line with our present QTD-analysis of the INS experiments, since it emphasizes the importance of the bath being a quantum system participating actively to the system-bath dynamics.

The analysis of reference [30] considers the QND measurement of a system *S* by a second system *D*, a detector. [*D* corresponds to the neutron, in our INS case]. $H_{SD}(t)$ is the time-dependent interaction Hamiltonian. Let the total "system + bath" Hamiltonian be

$$H_{tot} = H_S + H_B + H_{SB} \tag{12}$$

($H_S$: system free Hamiltonian; $H_B$: bath Hamiltonian; $H_{SB}$: system-bath interation). Including also the coupling of $H_{tot}$ with the detector *D*, one has:

$$H(t) = H_{tot} + H_{SD}(t) \tag{13}$$

The authors consider the evolution during an ultra-short time interval. In this context, it is crucial that the two coupling Hamiltonians $H_{SB}$ and $H_{SD}$ are not "simplified" to the common rotating-wave approximation of quantum optics [32]. That is, strict energy conservation between the system and the bath or the detector is not imposed [30]. [Recall that also in our INS case, energy conservation does not hold for "H + neutron", as repeatedly was pointed out]. Furthermore, the authors showed that if the detector disturbs impulsively the system, immediately after the measurement this action increases the system's energy (which is called "heating") [30]. This analytical result was confirmed by accompanying numerical simulations.

Let $\tau$ be the ultra-short time of the (nearly impulsive) quantum measurement (i.e., $\tau \to 0$) of the considered two-level system. In more formal terms, and for $0 \leq t \leq \tau$, the authors derived the following results:

$$\langle H_{SB}(0) \rangle \leq 0 \quad \to \quad \langle H_{SB}(\tau) \rangle = 0 \tag{14}$$
$$\langle H_{SD}(t) \rangle = -\langle H_{SB}(t) \rangle$$
$$\langle H_D \rangle = \text{const} \tag{15}$$

The energy transfers captured by these relations are associated with changes of the amount of entanglement (and other quantum correlations) between system and bath. This furthermore leads to redistribution of energy and entropy between the different parts of the complete system [30].

Further details of this process are as follows. At sufficiently short times $t = \tau + \Delta t$ after the measurement, in which holds $H_{SD}(t \geq \tau) = 0$, the common rotating wave approximation does not apply, which implies that $\langle H_S + H_B \rangle$ is not conserved, although, of course, $\langle H_{tot} \rangle$ is conserved. $\langle H_{SB} \rangle$ decreases towards its undisturbed value. Hence the following results hold:

$$\frac{d}{dt}(\langle H_S + H_B \rangle)\rfloor_{\tau + \Delta t} > 0 \tag{16}$$
$$\frac{d}{dt}\langle H_{SB} \rangle\rfloor_{\tau + \Delta t} < 0$$

These relations are equivalent to a quantum heating of the "system + bath" part [30], immediately after the disturbance.

The inverse effect was shown to happen a little later, i.e., one predicts a quantum cooling, which should be of interest for various experimental investigations, e.g., those on quantum gas condensates.

In connection with our INS experimental context, it should be observed that the neutron-H scattering process does not represent a QND measurement. Therefore, it should be mentioned that the above results have been extended to the general case of any arbitrary abrupt disturbance [31]. This special case is then applicable to the INS experimental context, as the neutron-H collision does represent such an abrupt disturbance.

For the INS examples of Section 4, the neutron-H collision corresponds to the abrupt measurement process (recall that the neutron corresponds to *D*), in which the system-bath (that is, the $H_2$-nanotube)

interaction causes a quantum heating (which corresponds to an anomalously increased $H_2$ translational energy). In the latter process, the quantum environment (i.e., the C-nanotube fragment adjacent to $H_2$) plays a significant role.

## 6. Thermodynamic Meaning of Negative Conditional Entropy, Generalized Landauer's Principle, and Interpretation of the INS Effect

In quantum-theoretical as well as information-theoretical discussions of Quantumness of Correlations, the concept of conditional entropy is of central importance; see, e.g., [5,14].

Consider two physical (resp. information-theoretical) systems $A$ and $B$, and let $S(A)$ and $S(B)$ be their well-known physical [33] (resp. Shannon [34]) entropies; cf. [5]. Furthermore, the combined (or complete) system $AB$ should have entropy $S(AB)$. Bayes' rule of classical probability theory introduces then the conditional entropy

$$S(A|B) = S(AB) - S(B) \tag{17}$$

which is a measure of our ignorance of $A$, under the condition that we have knowledge of $B$.

In classical theory, for the conditional entropy holds necessarily $S(A|B) \geq 0$. But in quantum theory, one easily shows that this is not always true [12]. For long time, this fact represented an interpretational problem. In the frame of quantum information theory, however, the negativity of conditional entropy was recently given an information-theoretical meaning with the aid of the quantum procedure known as quantum state merging; see, e.g., the review article [14]. But the meaning of it in the context of physical and/or thermodynamical processes remained unclear.

The situation changed with the original work of del Rio et al. [13]. Analyzing the standard implications of the "classical" Landauer's principle [35], the authors arrived to the conclusion that these implications no longer remain valid if quantum information is present; cf. also [36,37]. In the quantum case, the main finding is as follows: The work cost of information erasure is now determined by the quantum entropy of the system conditioned on the quantum information (of the system) being available to the observer (or to an observing Maxwell's demon)—the more an observer knows about the system, the less are the energy costs of the erasure under consideration. As one easily sees, this result emphasizes again the significance of conditional entropy of information, and thus it is fully in line with the well-known Landauer's statement: "Information is physical" [6].

In more formal terms, the main result of reference [13] may be concisely summarized as follows. Consider a system $A$ being correlated with a quantum memory $Q$. Then, the work that an observer (or demon) having access to $Q$ needs in order to carry out the erasure of the information content of $A$, is denoted by $W(A|Q)$. In the thermodynamic limit—which here means that the observer has access to many identical copies of $A$, and erases them jointly—the authors constructed a specific erasure procedure which has work costs not larger than

$$W(A|Q) = k_B T \ln(2) \, S(A|Q) \tag{18}$$

per single copy of $A$ [13]. $S(A|Q)$ is the well known von Neumann conditional entropy [14],

$$S(A|Q) = S(AQ) - S(Q) \tag{19}$$

Furthermore, assuming the validity of Landauer's principle for classical observers (demons), then these work costs are shown to be optimal.

Note also that Equation (18) represents a generalization (or "reformulation", see [13]) of the well known Landauer's principle of classical information theory

$$W(A) = k_B T \ln(2) \, H(A) \tag{20}$$

with $H(A) = -\mathrm{Tr}[\rho \log_2(\rho)]$.

However, for a quantum observer (or quantum Maxwell's demon), the situation may change significantly. For instance, as the last equation shows, now the work cost for erasure may be negative—i.e., if the observer has access to a quantum memory $Q$, the erasure process can even produce work (instead of consuming work).

This is rather easily shown with the aid of the following simple example. e.g., let the combined system $A + Q$ be a closed system in a pure quantum state, and thus its von Neumann entropy is zero,

$$S(AQ) = 0 \tag{21}$$

But then the reduced state of $Q$ is mixed, having a positive entropy $S(Q) > 0$. Hence Equation (19) yields $S(A|Q) < 0$, and therefore Equation (18) implies negative work costs,

$$S(A|Q) < 0 \quad \Rightarrow \quad W(A|Q) < 0 \tag{22}$$

which of course is tantamount to energy gain.

The "strange" negative conditional entropy gains with this result a thermodynamic operational meaning. This interpretation also extends earlier investigations dealing only with information-theoretical interpretations of this negativity (like, e.g., an enhanced quantum-communication resource); see related remarks and references in [13,14].

Moreover, as the authors of [13] noticed, this result shows that quantum discord can properly quantify the difference between the erasure-costs by using a quantum or classical memory $Q$.

It may be emphasized that these results do not violate the Second Law. Namely, the proposed process of erasure is not cyclic, and the negative work costs correspond to consumption of entanglement between the two quantum systems $A$ and $Q$. Crucially, if one wants to construct a cyclic process, another process is needed that restores the consumed entanglement (or quantum discord), which will have positive work costs. The conclusion was that the overall work costs of a cyclic process will be non-negative; see Addendum of reference [13].

### 6.1. Interpretation of the E-Excess INS Effect

Now, the considered "anomalous" effect can also be easily understood by referring to the considerations of the above Section 3.2, in particular with the aid of Equation (5).

Including explicitly the additional term

$$E_{int} \equiv E_{H,\mathcal{E}} > 0$$

due to the H-environment interaction in the energy conservation of the INS process,

$$\frac{\hbar^2 K_H^2}{2M_{eff}} = E - E_{H,\mathcal{E}} < E \tag{23}$$

we saw that this straightforwardly explains the well known conventional-theoretical increase of the H-effective mass,

$$M_{eff} > M_H$$

This was due to the reduced amount of energy, $E - E_{H,\mathcal{E}}$, available as kinetic energy to the struck H; see Section 3.

In the frame of QIT and Landauers's principle, this interaction energy may be also considered as the positive work costs for the erasure of classical bits happening by the INS process, when the struck H overcomes the binding potential and starts translation with kinetic energy $\hbar^2 K_H^2/2M_{eff}$.

But in the case of negative work costs, Equation (22), considered above, exactly the same consideration yields $E_{H-\mathcal{E}} < 0$ and hence

$$\frac{\hbar^2 K_H^2}{2M_{eff}} \equiv E + |E_{H,\mathcal{E}}| > E \tag{24}$$

with a decreased effective mass of H

$$M_{eff} < M_H = 1\,\text{a.m.u} \tag{25}$$

## 7. Quantum Correlations in Scattering Dynamics

After the above remarks and comments on the significance of QoC for the INS context, it is appropriate to consider in some more detail the time-dependence of the correlations accompanying the INS process.

Obviously, long time before the INS collision we have:

$$t = -\infty \quad : \quad \rho^n \otimes \rho^{H,\mathcal{E}} \tag{26}$$

The H-$\mathcal{E}$ interaction leads to mixed states for H and for $\mathcal{E}$). Let $\rho_\lambda^H$ be one of the H-states after the INS collision (with $\lambda$ indicating possible available states), and $\rho_\lambda^n$ be the associated mixed neutron state. Due to the H-$\mathcal{E}$ interactions and the fragility of entanglement, a fully entangled state for "neutron + H" does not exist. But then we may assume that a quantum discordant state exists, which is:

$$0 \approx t \leq \tau_{exp} \sim 10^{-14}\text{s} \quad : \quad \left( \int d\lambda\, p(\lambda)\, \rho^n(\lambda) \otimes \rho^H(\lambda) \right) \otimes \rho^{\mathcal{E}} \tag{27}$$

Here, $\tau_{exp} > 0$ is the very-short neutron-H scattering time window (roughly being in the pico- to femto-second range) due to the delta-like Fermi pseudo-potential [8]. The neutron-H interaction creates quantum correlations (in some exceptional cases, also are extremely short-lived transient neutron-H entanglement). These are represented by $\rho^{n,H}$. In simpler terms one may say that $\mathcal{E}$—or a Maxwell's demon associated with the environment—"observes" continuously the neutron-H system, thus causing a continuous change of the quantum correlations between neutron and H.

These H-$\mathcal{E}$ interactions will cause strong and complicated time-dependent quantum correlations, during some relaxation time $\tau_{relax}$, which is a characteristic quantity of the material at the existing temperature and pressure. Then, for times $\tau_{exp} < t \sim \tau_{relax}$, the complete system "neutron + H + $\mathcal{E}$" should be in a state described by $\rho^{n,H,\mathcal{E}}$:

$$\text{excited complete system}: \quad \tau_{exp} < t \sim \tau_{relax} \quad : \quad \rho^{n,H,\mathcal{E}} \tag{28}$$

At times $t \gg \tau_{relax}$, the neutron will be far away from the scattering system, and thus the system "H + $\mathcal{E}$" will evolve towards its earlier equilibrium. Thus we may write:

$$t \gg \tau_{relax} \quad : \quad \sum p_i \rho_i^{H,\mathcal{E}} \otimes \rho_i^n \tag{29}$$

(here, the sum can be replaced by an integral, and $p_i$ are classical probabilities). Each term in this discordant state is qualitatively similar to the correlated state of "H + $\mathcal{E}$" before the collision with the neutron. At the same time, the neutron occupies a mixed state $\rho_i^n$.

Later on, assuming an ideal high-resolution detector which measures momentum, and according to the well-known reduction postulate, an actual neutron detection will produce a pure neutron state and an associated quantum correlated "H + environment" state:

$$\text{detector measures } n \text{ at } t = +\infty \quad : \quad \rho_{\mathbf{k}_1}^{H,\mathcal{E}} \otimes |\mathbf{k}_1^\theta\rangle\langle\mathbf{k}_1^\theta| \tag{30}$$

This is the same assumption as conventional theory does—i.e., the plane wave approximation of conventional theory [8].

For simplicity of notation, $t$ is not shown explicitly in any time-dependent quantity appearing in several of the above formulas, like, e.g., $\rho^{H,\mathcal{E}}$.

From these considerations, it becomes obvious that the amount of quantum discord, and of any other measure of quantum correlations being relevant in this INS process, changes with time—e.g., in the course of time, quantum discord between any two of the three parts of "neutron + H + $\mathcal{E}$" may be erased and/or created continuously.

## 8. Excess *E*-Transfer in Neutron-H Collision—Number of "Consumed" Qubits

Moreover, here we can make contact of the theoretical frames of QTD and QIT with the INS results of the experimental part, Section 4, in particular with the *E*-excess "anomaly" under consideration.

To be specific, let us consider the roto-translational data of the $H_2$-nanotube system, (Figure 3, at $T = 23$ Kelvin). For example, at the chosen wavevector $K \equiv K_{tr} \approx 6$ Å$^{-1}$, the associated non-conventional *E*-surplus is $\Delta E_{tr} \approx 30$ meV—as obtained from the Figure by visual inspection. (i.e., the center of the experimental roto-recoil (translational) data, at this $K$ value, is ca. 30 meV higher than the corresponding point of the conventional recoil parabola for $M = 2$ a.m.u)

Here we associate this "anomalous" $\Delta E_{tr}$ with a corresponding value of negative conditional entropy in the generalized Landauer's principle, Equation (18). Then, for $K = 6$ Å$^{-1}$, we obtain the numerical estimate

$$\text{H}_2 \text{ in C-nanotube:} \quad \# \text{qubits} \sim \frac{\Delta E_{tr}}{k_B T \ln(2)} \sim 22 \tag{31}$$

$k_B T \ln(2)$ is Landauer's energy cost for erasure of one bit. In the presently considered case, this energy cost is negative.

In should be emphasized that this rough estimate is based on the (simplifying!) assumption that the quantum observer operates "fully efficiently" on the H-$\mathcal{E}$ system;, i.e., he extracts "optimally" the mentioned amount of "anomalous" excess *E*-transfer, which he then "transfers" to H. Furthermore, another assumption here is that no additional classical bits are "consumed" in this process, which would be related with positive costs of erasure. Hence, this estimated number of qubits should better be consider as representing an estimated rough lower bound of the qubit-numbers (which are being consumed with negative work costs) in the striking INS-effect under consideration. It may be intuitively helpful to compare these considerations with those of Section 6.1.

A crucial physical aspect to point out here is the fact that Landauer's principle and related quantum extensions are based on the conceptual framework of quasi-equilibrium statistical physics. However the ultafast INS process takes place far away from thermal equilibrium of the involved physical systems. This implies that the "temperature" $T$ appearing in the equations cannot be the common "equilibrium temperature" of thermodynamics; rather, it should represent a (still unknown) "effective temperature" $T^*$ characterizing the highly excited, short lived, and non-equilibrated, collisional system "neutron + H + $\mathcal{E}$". That is, in general $T \neq T^*$. Consequently, the above estimated number of consumed qubits will change with the choice of the "correct" temperature value, and thus, instead of the above estimate (31), one may expect that

$$\# \text{qubits} \sim 22 \, \frac{T}{T^*} \tag{32}$$

In this context, note that, in the terminology of del Rio et al. [13], the considered INS process represents an impulsive single-shot process, which is associated with a single instance of erasure. For such processes, reference [13] provides additional (and more general) theoretical results. We may hope that these results can shed more light on the physical properties and/or meaning of the effective temperature $T^*$.

With increasing energy and momentum transfers, also the considered "anomaly" $\Delta E_{tr}$ and the estimated number of involved qubits increase. In more illustrative terms, we can say that the quantum Maxwell's demon observing the scattering dynamics has more qubits to his disposition at higher *E*-transfers. This is also intuitively understandable due to the following simple observation: With increasing *E*-transfer (and during any given time interval characterizing the INS process) the translating $H_2$ interacts with a larger part of the nanotube-surface.

## 9. Additional Remarks and Discussion

Some additional considerations provided below may contribute to the clarification of the preceding theoretical and experimental results. It may be pointed out that only non-relativistic theory has been considered in this paper.

In view of the striking INS findings considered above, one may ask in which other potential systems similar scattering anomalies could be observed. Indeed, another interesting system is given by single $H_2O$ molecules in oriented nanoscale channels of a solid material, as firstly presented and analyzed in reference [17]; see also [18]; here, a large "anomaly" (about $-45$ %) of the effective mass of H in the OH-stretching vibrational motion was found. Moreover, several more such systems are already known, which however exhibit significantly smaller anomalies, and therefore they are less convincing., e.g., the same effect was originally found (in the electron-Volt energy-transfer range) in liquid $D_2$ [38], and in liquid $H_2$ [39]. Additionally, also scattering from H of solid polyethylene—a solid polymere, $(\text{-CH}_2\text{-})_n$—was found to exhibit the same small anomaly; see, e.g., [26]. The corresponding anomalies were in the range of a few percent, and thus they could be experimental artifacts caused by a wrong calibration of the spectrometer; for a related discussion, see Appendix A.

These observations lead to the following critical remarks. Although we showed that all usual, classical mechanisms cannot explain the INS anomalous results, coming with a plausible explanation based on the negative conditional entropy does not make it necessarily the cause of the observed discrepancies between INS experiments and conventional theory. From that point of view, one needs some explanation of why this happens in the particular INS problems considered above. Therefore, here it seems appropriate to stress the following point: The succinct specifics of the studied materials that give rise to the considered "anomalous" quantum correlation effects are not known until now. These specifics certainly concern, or are related with, the specifics of the H-$\mathcal{E}$ quantum-correlation dynamics (see Section 7), which however are beyond the reach of present-day quantum chemistry of H-$\mathcal{E}$ interactions—the latter dealing with quantum statics (i.e., time-independent problems), rather than with quantum dynamics (i.e., time-dependent processes).

All considerations and results in this paper are related to concrete INS experiments, which is incoherent. In simple terms, "incoherent" means that the impinging neutron collides with, and scatters from, a single particle (nucleus, atom, etc.). In the case of neutron scattering from protons (conventionally also referred to as H-atoms), the scattering is incoherent in a very good approximation, due to the spin-flip mechanism accompanying the neutron-proton collision [8]. This is tantamount to the gain of a "which-path information" [32] which destroys coherence between possible neutron paths. A clear first-principles explanation of this effect, and of coherent versus incoherent scattering, is presented in the well-known "Feynman Lectures" [7], in particular Section 3.3 therein.

Contrary to the viewpoint of this paper, in standard neutron scattering theory, quantum correlations between distinguishable particles are assumed to play no role; cf. [8,10,11]. Namely, although neutron is a quantum particle and, in general, will become even entangled with a scattering nucleus, this quantum effect is absent in the standard theory. Moreover, the neutron's quantum degrees-of-freedom are absent in any basic scattering expression or formula (e.g., the dynamic structure factor; or the van Hove correlation functions). Instead, only *c*-numbers referring to a neutron appear in these formulas. A nearer consideration of this issue shows that this "absence" is not due to some mathematical "error", but only due to the simple fact that conventional theory is based on the formalism of first-order perturbation theory. In this approximation, the degrees of freedom of neutron

and H (or any other nucleus) are formally decoupled; so no neutron-H quantum correlation effects do appear here.

In other terms, text-book non-relativistic neutron scattering theory treats the neutron as a classical mass point [8,10,11]; i.e., this theory contains only *c*-numbers referring to neutron's properties (e.g., the scattering length $b_A$ or the cross-section $\sigma_A$).

The excess energy transfer, and the associated effective-mass reduction of the scattering particle *H* in the presented INS experiments, are sometimes considered as violating the energy and momentum conservation laws of basic physics. However, this is not the case because *H*, and also "neutron + H" are *not* closed quantum systems, but open systems. It is helpful to write down explicitly the conservation relations for the complete closed system "neutron + H + $\mathcal{E}$ (environment)"

$$E_{H+\mathcal{E}} = -E_n \quad \text{and} \quad \hbar K_{H+\mathcal{E}} = -\hbar K_n, \tag{33}$$

which express the correct energy and momentum conservations for a closed system. Obviously, these are in contrast to the following (often assumed) equations

$$E_H = -E_n \quad \text{and} \quad \hbar \mathbf{K}_H = -\hbar \mathbf{K}_n, \tag{34}$$

which however are (in various cases only crude) approximations.

Clearly, if one neglects $\mathcal{E}$ (or over-simplifies its quantum character), the conservation laws seem artificially violated. Thus we can say that the quantum dynamics of $\mathcal{E}$ is indispensable for the interpretation of the considered INS effect from H.

In the light of these remarks, the following (for conventional theory: highly perplexing) conclusion follows as a corollary: The time evolution of any neutron-H dynamical process (like, e.g., INS), in which H belongs to a hydrogenous material (and thus one cannot neglect $\mathcal{E}$), is not unitary.

The striking INS results from $H_2$ in C-nanotubes [28] and metal-organic materials [29], discussed in Section 4, may appear contradictory because: (i) the $J = 0 \rightarrow 1$ rotational excitation reveals the conventional value $M_{eff} \approx 1$ a.m.u., see Equation (10). But (ii) the $M_{eff}$ of the translational motion of $H_2$ is not 2 a.m.u. as conventionally expected, but only $M_{eff} \approx 0.64$ a.m.u. However, this "contradiction" is an artifact that just disappears in the light of the QTD theoretical treatment. Namely, this difference only implies that the quantum environments of H are qualitatively different in these two cases. In other words: the QTD theoretical frame allows us to reveal novel quantum effects of "system-environment" interactions that are beyond the predictive power of present-day quantum chemistry.

It should be emphasized that both "system" and "environment" represent quantum objects, the quantum dynamics of which must be treated on equal footing. In simple words, environment is not a passive object that just restricts (or confines) the system's dynamics.

Based on the INS experimental effect under consideration (and in particular of the striking reduction of effective mass of H), a speculative idea arises, which concerns the possible practical and/or technological relevance of QTD and quantum information theory: A more mobile H atom, or $H^+$ ion, due to a suitably chosen nanostructured fuel cell material, would cause an improved cell's efficiency.

Last but not least, it should be mentioned that the considered "anomalous" INS effect has recently found an additional theoretical interpretation [26,40] based on "quantum mechanisms" (rather than on quantum correlations), which is based on the fundamental theory of Weak Values (WV) and Two-State Vector Formalism (TSVF) by Aharonov and collaborators, cf. [41,42] and the extended list of publications on WV-TSFV cited in [26]. In this theoretical frame, the striking effect under consideration is best described as a momentum deficit effect [26,40].

The prospect (or perhaps, speculative idea) to apply concepts of QIT in establishing connection of it with "experimental INS from H"—as done in Section 6.1. and Section 8—may motivate further related explorations and shed more light on the physics, chemistry and technologies of H-containing materials.

In conclusion, it seems that the theoretical frameworks of quantum thermodynamics and quantumness of correlations, in combination with modern scattering and quantum-optical experimental techniques, shed new light on interpretational questions concerning fundamental quantum theory—like, e.g., the time-inversion invariance of the basic physical laws, the meaning of the wavefunction, operational meaning of entanglement and quantum discord, etc. Moreover, they also offer a qualitatively new guide to our intuition in predicting new effects and revealing new quantum phenomena, which promotes multidisciplinary scientific and technological research.

**Funding:** Part of this research was funded by European COST Action MP1403 (Nanoscale Quantum Optics).

**Acknowledgments:** I wish to thank Philipp Stammer (Max-Born-Institute Berlin and TU Berlin) for helpful discussions.

**Conflicts of Interest:** The author declares no conflict of interest.

## Abbreviations

The following abbreviations are used in this paper:

| | |
|---|---|
| a.m.u. | Atomic Mass Unit |
| FSE | Final-State Effects |
| IFM | Interaction Free Measurement |
| IA | Impulse Approximation |
| IINS | Inelastic Incoherent Neutron Scattering |
| INS | Equivalent to IINS |
| IT | Information theory |
| meV | Milli-Electron Volt |
| MZI | Mach–Zehnder Interferometer |
| NP | nondeterministic polynomial |
| QE | Quantum Entanglement |
| QTD | Quantum Thermodynamics |
| QIT | Quantum Information Theory |
| QoC | Quantumness of Correlations |
| QND | Quantum Non-Demolition |
| TOF | Time-of-Flight |
| TSVF | Two-State Vector Formalism |
| WV | Weak Value |

## Appendix A. Experimental Context—Incoherent Neutron Scattering

It appears that instrumental details are crucial for evaluating applications of the theoretical QTD- and QIT-frame under consideration, because they show explicitly "what is directly measured", i.e., independently from individual theoretical assumptions and/or beliefs.

*Appendix A.1. Incoherent Inelastic Neutron Scattering from Protons*

Newest generation neutron scattering spectrometers are time-of-flight (TOF) instruments; cf. Figure A1. Here, a short pulse of neutrons reaches the first monitor of the spectrometer, which triggers the measurement of TOF. Subsequently, a neutron scatters from the sample and may reach the detector, which stops the TOF measurement. The determined values of momentum and energy transfers (which are the variables of the dynamic structure factor $S(K, E)$ are not directly measured (as one often believes); see below.

Modern neutron spectrometers, like, e.g., ARCS [15] (at Spallation Neutron Sourse SNS, Oak Ridge Nat. Lab., USA) have several thousand individual detectors (detector pixels). From each TOF-value measured with each individual detector at scattering angle $\theta$, the associated transfers of momentum ($\hbar K$) and energy ($E = \hbar\omega$) from the neutron onto the struck particle (e.g., a H atom) are uniquely determined; see, e.g., [18,26]. As shortly explained below, an individual detector measures along one

specific trajectory in the whole $K - E$ plane. (The latter notation is commonly used, instead of the more precise expression $\hbar K - E$.)

For a self-contained and detailed presentation of the TOF experimental procedure, see [18,26].

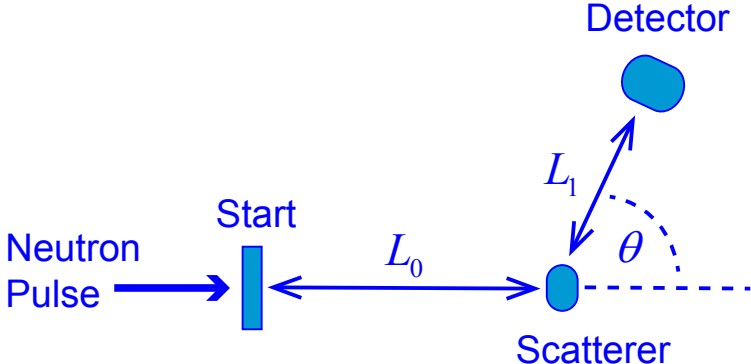

**Figure A1.** Scheme of a time-of-flight neutron scattering setup. (Taken from [26], with permission of Quanta).

*Appendix A.2. Details of Scattering Spectrometer, Calibration and "What Is Measured"*

An outline of a standard TOF scattering experiment is as follows; cf. Figure A1. A short pulse of neutrons produced by the spallation source reaches the monitor, (i.e., the clock that starts the TOF measurement). $L_0$ is the distance between monitor and sample. A neutron of the pulse that scatters from the sample reaches the detector, which triggers the stop signal of the TOF measurement. $L_1^\theta$ (or $L_1$, for simplicity) is the sample–detector distance; the corresponding scattering angle is $\theta$. Here we consider isotropic scattering only.

For a measured TOF value $t_{TOF}$ holds

$$t_{TOF} = \frac{L_0}{v_0} + \frac{L_1^\theta}{v_1} + t_0 \tag{A1}$$

$v_0$ and $v_1$ are the velocities of the neutron before and after scattering, respectively; $t_0$ is a time-offset parameter caused by effects of the electronics. When $v_0$ and thus also the initial kinetic energy $E_0$ have experimentally predetermined fixed values, one speaks of a "direct geometry TOF-spectrometer". ARCS and MARI, which are mentioned in the present paper, belong to this class of spectrometers.

The scattering intensity measured by the detector is a function of the neutron's energy transfer (or, energy loss)

$$\begin{aligned} E &= E_0 - E_1 = \frac{1}{2}mv_0^2 - \frac{1}{2}mv_1^2 \\ &\equiv \hbar\omega = \frac{(\hbar k_0)^2}{2m} - \frac{(\hbar k_1)^2}{2m} \end{aligned} \tag{A2}$$

($m$: neutron mass) and the corresponding momentum transfer on the neutron

$$\hbar\mathbf{K}_n = \hbar\mathbf{k}_1 - \hbar\mathbf{k}_0 \tag{A3}$$

(Bold letters represent vectors.) It holds

$$|\mathbf{K}_n| = K_n = \sqrt{k_0^2 + k_1^2 - 2k_0 k_1 \cos\theta} \tag{A4}$$

where $k_i$ is the modulus of vector $\mathbf{k}_i$. Since we consider here incoherent [7,8] scattering, the exchange of momentum and energy is between a neutron and (the nucleus of) a single atom [7,8]. Due to Newton's "actio = reactio", in strict two-body scattering the struck atom (nucleus) receives the momentum transfer

$$\mathbf{K}_A = -\mathbf{K}_n \tag{A5}$$

and one defines

$$K \equiv |\mathbf{K}_A| = |\mathbf{K}_n| \tag{A6}$$

Assuming that the sample is initially at rest, the neutron scatters from an atom with mass $M$ and zero mean initial momentum, i.e., $\langle P \rangle = 0$. Here, assuming conservation of kinetic energy and momentum in a binary collision, it holds the simple kinematic relation

$$\frac{k_1}{k_0} = \frac{\cos\theta + \sqrt{(M/m)^2 - \sin^2\theta}}{M/m + 1} \tag{A7}$$

Parenthetically, the shape analysis and theoretical interpretation of a measured intensity peak constitutes a large part of many neutron scattering investigations.

The following facts should be pointed out.

(a) The measured energy and momentum transfer values, which then are needed for the theoretical interpretation of the experimental data, are those of the neutron—and not those of the struck H, which are entering the dynamical structure factor $S(K, E)$.

(b) From the measured TOF value (A1) follow the values of $v_1$, Equation (A1), of $k_1 = |\mathbf{k}_1|$, and consequently of energy transfer $E = \hbar\omega$, see Equation (A2). Note that the value of $\theta$ plays no role here. The value of $E$—for each single elementary scattering process—is independent of any tacit theoretical beliefs or assumptions, because it follows from the actually measured TOF value immediately.

(c) Momentum transfer $\hbar\mathbf{K}$, Equation (A4), is determined from both the measured TOF value and the direction given by $\theta$ in which the detector is positioned. However, the direct experimental determination of $\theta$ (which is given by the geometry of the spectrometer) is by no means an easy matter. Furthermore, one uses Equation (A7) which connects the value of $\theta$ and the atomic mass $M$. In other words, one takes for granted that the atomic mass of $M$ is well known (recall that non-relativistic quantum mechanics holds here), and thus the conventionally expected ratio $k_1/k_0$ is fully determined.

The following point is crucial: If the scattering particle is not free but "dressed" with environmental degrees of freedom, then the quantity $M$ appearing in the above kinematic equations will be different from that of the free particle. This is crucial to note, as it is quite often a common practice—at least in calibration of spectrometers for neutron Compton scattering (NCS) or deep-inelastic neutron scattering (DINS)—to use the free-mass $M$-value in order to "fine-tune" and "improve" the calibration of the instrument, i.e., to "optimize" the instrumental values of $\theta$ and/or $L_1$, aiming to achieve an agreement of the measured scattering data with conventional theory; for a detailed example, see [38] This is often done using Equation (A7) on scattering data from H, since the H-neutron angular scattering dependence is very sensitive to the $\theta$-value, as Equation (A7) shows; for a detailed discussion see [38].

The common justification for doing so is along these lines: The impulsive neutron-nucleus interaction is very short, due to the small range of the Fermi pseudo-potential of the strong force, and thus a neutron only collides with a nucleus being "effectively free".

As a matter of fact, this procedure is based on the belief that the simple (and strict) two-body elastic collision is a fully valid assumption, and thus the free-atom mass is the correct quantity for the parameter $M$ entering the kinematic Equations (A1)–(A7). Hence, to investigate the new effect of neutron-H scattering under consideration, a fair procedure for determination of $\theta$ (and any other

instrumental parameter) is needed which is "independent" of theoretical beliefs and/or assumptions; for a detailed analysis see [38].

Summarizing, from each measured $t_{TOF}$ in direction $\theta$, the corresponding momentum ($\hbar K$) and energy ($E = \hbar\omega$) transfers of the neutron are uniquely determined. Thus we realize that each specific detector measures one specific trajectory, or a narrow band, in the two-dimensional $K - E$ plane.

In neutron scattering experiments from condensed systems, the struck (recoiling) particles of the sample cannot be measured.

The scattering process produces an outgoing three-dimensional wave for the entangled atom-neutron system, which, for amorphous samples, is isotropic, since the neutron's de Broglie wavelength is much larger than the diameter of the nucleus [8]. This is the so-called *s*-wave scattering [8].

According to Equations (A2) and (A4), the detector's instrumental parameters {IP} shown in Equation (A1) determine the corresponding scattering intensity $I(\mathbf{K}, E)$ and the corresponding dynamical structure factor $S(\mathbf{K}, E)$ of the scattering system. The latter is of central importance for the theoretical analysis of experimental data [8]. In short:

$$I(\text{TOF}, \{\text{IP}\}) \;\; \Rightarrow \;\; I(\mathbf{K}, E) \;\; \Rightarrow \;\; S(\mathbf{K}, E) \tag{A8}$$

However, the following fact cannot be overemphasized: As they stand, these relations can be ambiguous and/or misleading, because the experimental quantities $\mathbf{K}$ and $E$ are those of the neutron, and not of the scattering particle, i.e., H, in our INS case. That is, one still needs to determine $\mathbf{K}_H$ and $E_H$.

Thus, the schematic relation (A8) is in general not correct, because $S(\mathbf{K}, E)$ of the scattering sample does not contain the quantities $\mathbf{K}_H$ and $E_H$ referring to the struck H, which are not directly measurable. Hence, one may obtain the desired dynamical structure factor $S(\mathbf{K}_H, E_H)$ of the sample under investigation only with some additional theoretical assumption, like, e.g., the conventionally done approximation

$$E_H \approx -E_n \quad \text{and} \quad \hbar\mathbf{K}_H \approx -\hbar\mathbf{K}_n, \tag{A9}$$

which express energy and momentum conservation only for a closed neutron-H system—which, in general, is not the case in our INS context because they ignore the role of the environment. See the main text for more explanations and discussion.

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
