# Peer review of "Experimental Implications of Negative Quantum Conditional Entropy—H2 Mobility in Nanoporous Materials"

_applsci, doi:10.3390/app10228266_

Round 1

Reviewer 1 Report

In his paper about experiments determining "H2 mobility in nanoporous materials", the author shows the
necessity to introduce the concept of negative quantum conditional entropy in marked contrast with the
classical case. This conclusion is obtained by comparing existing litterature and his own theory.
Quoting the author in the abstract ""the data of the H2 translational motion along the nanotube axis
seems to show that the neutron apparently scatters from a fictitious particle with mass of 0.64 atomic
mass units (a.m.u.) — instead of the value of 2 a.m.u., as conventionally expected." This discrepancy
is explained if one takes into account the negativity of the conditional entropy.

The paper is well documented and the also well written. It merits the publication.
In the attached document, I have highlighted some questions I have about the notation (in particular the use
of bold letters).

Author Response

I thank Reviewer 1 very much for this/her kind comments nd suppostive remarks,  and the many typos and missing definitions localized. All necessary changes have been done.

Reviewer 2 Report

This paper discusses an intriguing suggestion that negative quantum conditional entropy may explain anomalies in the inelastic neutron scattering (INS) from protons of H2 molecules in nanoscale environments. The paper provides all the context needed from both quantum thermodynamics and neutron scattering for the reader to apprehend the reasons why usual explanations cannot elucidate the INS data. In my opinion, this makes it very readable, even for non-specialists.

The paper makes a coherent case for why a negative quantum conditional entropy is a plausible explanation of the experimental observations of lighter than expected effective mass of the proton (0.64 a.m.u.). Although the author shows that all usual, classical mechanisms cannot explain the results, coming with a plausible explanation does not make it necessarily the cause of the discrepancy. From that point of view, it would have been nice if the author could provide some explanation of why this happens in this particular scattering problem. The paper does not really elaborate on the specifics of the studied systems that give rise to these unusual quantum correlations. Such a discussion would provide stronger support for the paper conclusions. For the same reasons, the author could elaborate on which other potential systems similar scattering anomalies could be observed. Also, it would be interesting if the author could discuss the meaning of the finding that the associated energy cost of erasure is ~22 qubits.

There are minor typos that could be corrected (found a few at lines 114, 139, the caption of Fig. 3, lines 263, 592). The list of abbreviations is incomplete (at least I.A. is missing).

Author Response

I thank Reviewer 2 very much for his/her insightful and conssttructive critical remarks,  and supportive comments.

Reviewer made (in 2nd paragraph of report) three major points. All of them have been addressed explicitly in the revised Manuscrript:

Point 1 in lines 395-405 ;  Point 2 in lines 384-394;  Point 3 in ca 10 lines around Equation 32.    All new inclusions/text are marked with YELLOW in the attached pdf-flile "Dreismann-marked-corrections.....pdf"

I also thank the Reviewer for his/her kind and supportive comments on 1st paragraph of the report.

The mentioned typos and missing abbreviations have been corrected/added.
